# Semi-supervised learning by selective training with pseudo labels via confidence estimation

## Abstract

We propose a novel semi-supervised learning (SSL) method that adopts selective training with pseudo labels. In our method, we generate hard pseudo-labels and also estimate their confidence, which represents how likely each pseudo-label is to be correct. Then, we explicitly select which pseudo-labeled data should be used to update the model. Specifically, assuming that loss on incorrectly pseudo-labeled data sensitively increase against data augmentation, we select the data corresponding to relatively small loss after applying data augmentation. The confidence is used not only for screening candidates of pseudo-labeled data to be selected but also for automatically deciding how many pseudo-labeled data should be selected within a mini-batch. Since accurate estimation of the confidence is crucial in our method, we also propose a new data augmentation method, called MixConf, that enables us to obtain confidence-calibrated models even when the number of training data is small. Experimental results with several benchmark datasets validate the advantage of our SSL method as well as MixConf.

## 1 Introduction

Semi-supervised learning (SSL) is a powerful technique to deliver a full potential of complex models, such as deep neural networks, by utilizing unlabeled data as well as labeled data to train the model. It is especially useful in some practical situations where obtaining labeled data is costly due to, for example, necessity of expert knowledge. Since deep neural networks are known to be "data-hungry" models, SSL for deep neural networks has been intensely studied and has achieved surprisingly good performance in recent works (Van Engelen & Hoos, 2020). In this paper, we focus on SSL for a classification task, which is most commonly tackled in the literature.

Many recent SSL methods adopt a common approach in which two processes are iteratively conducted: generating pseudo labels of unlabeled data by using a currently training model and updating the model by using both labeled and pseudo-labeled data. In the pioneering work (Lee, 2013), pseudo labels are hard ones, which are represented by one-hot vectors, but recent methods (Tarvainen & Valpola, 2017; Miyato et al., 2018; Berthelot et al., 2019; 2020; Verma et al., 2019; Wang et al., 2019; Zhang & Qi, 2020) often utilize soft pseudo-labels, which may contain several non-zero elements within each label vector. One simple reason to adopt soft pseudo-labels is to alleviate confirmation bias caused by training with incorrectly pseudo-labeled data, and this attempt seems to successfully contribute to the excellent performance of those methods. However, since soft pseudo-labels only provide weak supervisions, those methods often show slow convergence in the training (Lokhande et al., 2020). For example, MixMatch (Berthelot et al., 2019), which is one of the state-of-the-art SSL methods, requires nearly 1,000,000 iterations for training with CIFAR-10 dataset. On the other hand, in this paper, we aim to utilize hard pseudo-labels to design an easy-to-try SSL method in terms of computational efficiency. Obviously, the largest problem to be tackled in this approach is how to alleviate the negative impact caused by training with the incorrect pseudo-labels.

In this work, we propose a novel SSL method that adopts selective training with pseudo labels. To avoid to train a model with incorrect pseudo-labels, we explicitly select which pseudo-labeled data should be used to update the model. Specifically, assuming that loss on incorrectly pseudo-labeled data sensitively increase against data augmentation, we select the data corresponding to relatively

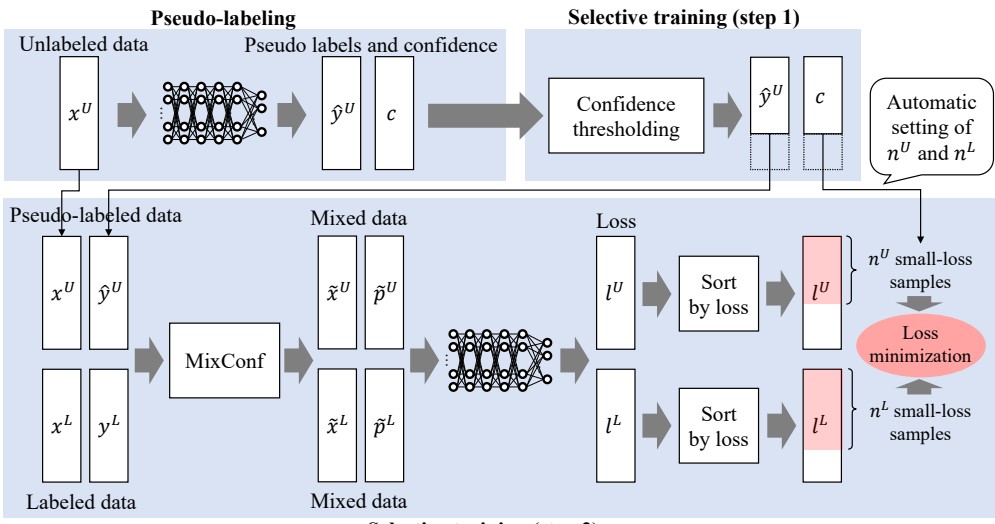

Figure 1: An overview of the proposed semi-supervised learning method.

small loss after applying data augmentation. To effectively conduct this selective training, we estimate confidence of pseudo labels and utilize it not only for screening candidates of pseudo-labeled data to be selected but also for automatically deciding how many pseudo-labeled data should be selected within a mini-batch. For accurate estimation of the confidence, we also propose a new data augmentation method, called MixConf, that enables us to obtain confidence-calibrated models even when the number of training data is small. Experimental results with several benchmark datasets validate the advantage of our SSL method as well as MixConf.

## 2 PROPOSED METHOD

Figure 2 shows an overview of our method. Given a mini-batch from labeled data and that from unlabeled data, we first generate pseudo labels of the unlabeled data based on predictions of the current model. Let $x \in \mathbb{R}^m$, $y \in \{1, 2, ...C\}$, and $f : \mathbb{R}^m \to \mathbb{R}^C$ denote input data, labels, and the classifier to be trained, respectively. Given the input unlabeled data $x^{\mathrm{U}}$, the pseudo label $\hat{y}^{\mathrm{U}}$ is generated by simply taking $\arg\max$ of the classifier's output $f(x^{\mathrm{U}})$. Then, we conduct selective training using both the labeled data and the pseudo-labeled data. In this training, to alleviate negative effect caused by training with incorrect pseudo-labels, we explicitly select which data should be used to update the model. Below, we describe details of this selective training.

### 2.1 SELECTIVE TRAINING WITH PSEUDO LABELS BASED ON CONFIDENCE

As described previously, the pseudo labels are generated based on the predictions of the current model, and we assume that the confidence of those predictions can be also computed in addition to the pseudo labels. When we use a popular architecture of deep neural networks, it can be obtained by simply taking $\max$ of the classifier's output (Hendrycks & Gimpel, 2016) as:

$$c_i = \max_{j \in \{1, 2, ..., C\}} f(x_i^{\mathrm{U}})[j], \tag{1}$$

where $c_i$ is the confidence of the classifier's prediction on the $i$-th unlabeled data $x_i^{\mathrm{U}}$, and $f(x)[j]$ is the $j$-th element of $f(x)$. When the model is sufficiently confidence-calibrated, the confidence $c_i$ is expected to match the accuracy of the corresponding prediction $f(x_i^{\mathrm{U}})$ (Guo et al., 2017), which means it also matches the probability that the pseudo label $\hat{y}_i^{\mathrm{U}}$ is correct.

To avoid training with incorrect pseudo-labels, we explicitly select the data to be used to train the model based on the confidence. This data selection comprises two steps: thresholding the confidence and selecting relatively small loss calculated with augmented pseudo-labeled data. The first step is

quite simple; we pick up the pseudo-labeled data that have higher confidence than a certain threshold $c_{\text{thr}}$ and discard the remaining. In the second step, MixConf, which will be introduced later but is actually a variant of Mixup (Zhang et al., 2018), is applied to both the labeled and unlabeled data to augment them. As conducted in (Berthelot et al., 2019), we shuffle all data and mix them with the original labeled and pseudo-labeled data, which results in $\{(\tilde{x}_i^{\text{L}}, \tilde{p}_i^{\text{L}})\}_{i=1}^{B_{\text{L}}}$ and $\{(\tilde{x}_j^{\text{U}}, \tilde{p}_j^{\text{U}})\}_{j=1}^{B_{\text{U}}}$, respectively, where $p \in \mathbb{R}^C$ is a vector-style representation of the label that is adopted to represent a mixed label. Then, we calculate the standard cross entropy loss for each mixed data. Finally, we select the mixed data that result in relatively small loss among the all augmented data, and only the corresponding small-loss is minimized to train the model.

Why does the small-loss selection work? Our important assumption is that the loss calculated with incorrect labels tends to sensitively increase when the data is augmented. This assumption would be supported by effectiveness of the well-known technique, called test-time augmentation (Simonyan & Zisserman, 2015), in which incorrect predictions are suppressed by taking an average of the model's outputs over several augmentations. Since we conduct the confidence thresholding, the loss corresponding to the pseudo-labeled data is guaranteed to be smaller than a certain loss level defined by the threshold $c_{\text{thr}}$. However, when we apply data augmentation, that is MixConf, to the pseudo-labeled data, the loss related to incorrectly pseudo-labeled data becomes relatively large, if the above assumption is valid. It means that selecting relatively small loss after applying MixConf leads to excluding incorrect pseudo-labels, and we can safely train the model by using only the selected data.

Han et al. (2018) and Lokhande et al. (2020) have presented similar idea, called small-loss trick (Han et al., 2018) or speed as a supervisor (Lokhande et al., 2020), to avoid training with incorrect labels. However, their assumption is different from ours; it is that loss of incorrectly labeled data decreases much slower than that of correctly labeled data during training. Due to this assumption, their methods require joint training of two distinct models (Han et al., 2018) or nested loop for training (Lokhande et al., 2020) to confirm which data show relatively slow convergence during training, which leads to substantially large computational cost. On the other hand, since our method focuses on change of loss values against data augmentation, not that during training, we can efficiently conduct the selective training by just utilizing data augmentation in each iteration.

Since the confidence of the pseudo label represents the probability that the pseudo label is correct, we can estimate how many data we should select based on the confidence by calculating an expected number of the mixed data generated from two correctly labeled data. Specifically, when the averaged confidence within the unlabeled data is equal to $c_{\text{ave}}$, the number of the data to be selected can be determined as follows:

$$n_{\text{L}} = \frac{B_{\text{L}} + c_{\text{ave}} B_{\text{U}}}{B_{\text{L}} + B_{\text{U}}} B_{\text{L}}, \tag{2}$$

$$n_{\text{U}} = \min\left(B_{\text{L}}, \frac{B_{\text{L}} + c_{\text{ave}} B_{\text{U}}}{B_{\text{L}} + B_{\text{U}}} c_{\text{ave}} B_{\text{U}}\right), \tag{3}$$

where $n_{\text{L}}$ is for the data generated by mixing the labeled data and shuffled data, and $n_{\text{U}}$ is for those generated by mixing the unlabeled data and shuffled data. Here, to avoid too much contribution from the pseudo-labeled data, we restrict $n_{\text{U}}$ to be smaller than $B_{\text{L}}$. Within this restriction, we can observe that, if we aim to perfectly balance $n_{\text{L}}$ and $n_{\text{U}}$, $B_{\text{U}}$ should be set to $B_{\text{L}}/c_{\text{ave}}$. However, $c_{\text{ave}}$ cannot be estimated before training and can fluctuate during training. Therefore, for stable training, we set $B_{\text{U}} = B_{\text{L}}/c_{\text{thr}}$ instead and fix it during training.

Finally, the total loss $\mathcal{L}$ to be minimized in our method is formulated as the following equation:

$$\mathcal{L} = \frac{1}{B_{\text{L}}} \sum_{i=1}^{n_{\text{L}}} l(\tilde{x}_{s[i]}^{\text{L}}, \tilde{p}_{s[i]}^{\text{L}}) + \lambda_{\text{U}} \frac{1}{B_{\text{L}}} \sum_{j=1}^{n_{\text{U}}} l(\tilde{x}_{t[j]}^{\text{U}}, \tilde{p}_{t[j]}^{\text{U}}), \tag{4}$$

where $l$ is the standard cross entropy loss, $s$ and $t$ represent the sample index sorted by loss in an ascending order within each mini-batch, and $\lambda_{\text{U}}$ is a hyper-parameter that balances the two terms.

To improve the accuracy of pseudo labels as well as their confidence, we can average the model's outputs over $K$ augmentations to estimate pseudo labels as conducted in (Berthelot et al., 2019). In that case, we conduct MixConf for all augmented pseudo-labeled data, which results in $K$ mini-batches each of which contains $B_{\text{U}}$ mixed data. Therefore, we need to modify the second term in the

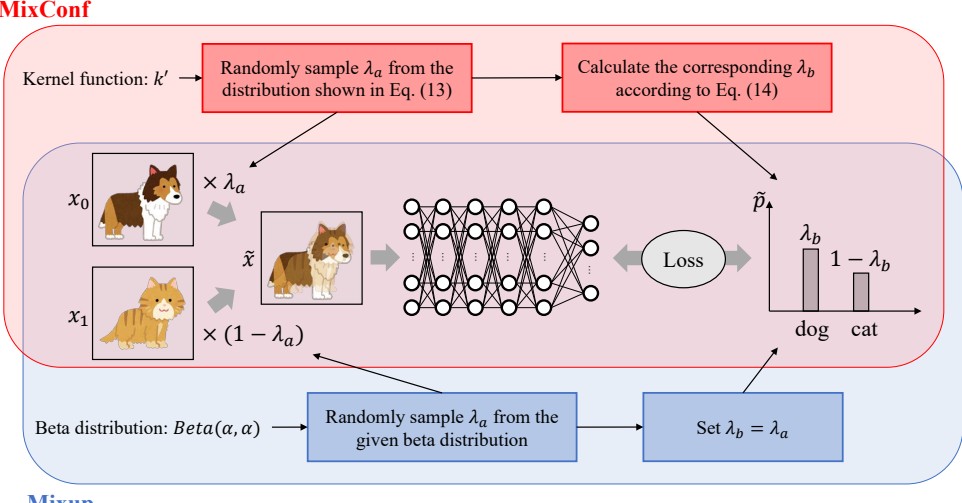

Figure 2: An overview of MixConf and its comparison with Mixup. MixConf basically follows the scheme of Mixup, but it carefully sets the interpolation ratios $(\lambda_a, \lambda_b)$ so that training with the interpolated data leads to better calibrated models. Unlike Mixup, $\lambda_b$ is not necessarily equal to $\lambda_a$.

right-hand side of Eq. (4) to take the average of losses over all the mini-batches. In our experiments, we used $K = 4$ except for an ablation study.

## 2.2 MIXCONF TO OBTAIN BETTER CALIBRATED MODELS

In the previous section, we assumed that the model is sufficiently confidence-calibrated, but deep neural networks are often over-confident on their predictions in general (Guo et al., 2017). This problem gets bigger in case of training with a small-scale dataset as we will show in our experiments. Consequently, it should occur in our SSL setting, because there are only a small amount of labeled training data in the early stage of the training. If the confidence is over-estimated, incorrect pseudo-labels are more likely to be selected to calculate the loss due to loose confidence-thresholding and over-estimated $(n_{\mathrm{L}}, n_{\mathrm{U}})$, which should significantly degrade the performance of the trained model. To tackle this problem, we propose a novel data augmentation method, called MixConf, to obtain well-calibrated models even when the number of training data is small. MixConf basically follows the scheme of Mixup, which is known to contribute to model's calibration (Thulasidasan et al., 2019), but is more carefully designed for confidence calibration.

Figure 2 shows an overview of MixConf. In a similar way with Mixup, MixConf randomly picks up two samples $\{(x_0, p_0), (x_1, p_1)\}$ from the given training dataset and generates a new training sample $(\tilde{x}, \tilde{p})$ by linearly interpolating these samples as the following equations:

$$\tilde{x} = \lambda_a x_0 + (1 - \lambda_a) x_1, \tag{5}$$
$$\tilde{p} = \lambda_b p_0 + (1 - \lambda_b) p_1, \tag{6}$$

where $\lambda_a \in [0, 1]$ and $\lambda_b \in [0, 1]$ denote interpolation ratios for data and labels, respectively. Note that $\lambda_a$ is not restricted to be equal to $\lambda_b$ in MixConf, while $\lambda_a = \lambda_b$ in Mixup. Since Mixup is not originally designed to obtain confidence-calibrated models, we have to tackle the following two questions to obtain better calibrated models by such a Mixup-like data augmentation method:

- How should we set the ratio for the data interpolation?
  (In case of Mixup, $\lambda_a$ is randomly sampled from the beta distribution)
- How should we determine the labels of the interpolated data?
  (In case of Mixup, $\lambda_b$ is set to be equal to $\lambda_a$)

We first tackle the second question to clarify what kind of property the generated samples should have. Then, we derive how to set $\lambda_a$ and $\lambda_b$ so that the generated samples have this property.

### 2.2.1 How to determine the labels of the interpolated data

Let us consider the second question shown previously. When the model predicts $\hat{y}$ for the input $x$, the expected accuracy of this prediction should equal the corresponding class posterior probability $p(\hat{y}|x)$. It means that, if the model is perfectly calibrated, its providing confidence should match the class posterior probability. On the other hand, from the perspective of maximizing the prediction accuracy, the error rate obtained by the ideally trained model should match the Bayes error rate, which is achieved when the model successfully predicts the class that corresponds to the maximum class-posterior probability. Considering both perspectives, we argue that $\max_j f(x)[j]$ of the ideally trained model should represent the class posterior probability to have the above-mentioned properties. Therefore, to jointly achieve high predictive accuracy and confidence calibration, we adopt the class posterior probability as the supervision of the confidence on the generated data. Specifically, we aim to generate a new sample $(\tilde{x}, \tilde{p})$ so that it satisfies $\tilde{p} = p(y|\tilde{x})$.

Although it is quite difficult to accurately estimate the class posterior probability for any input data in general, we need to estimate it only for the linearly interpolated data in our method. Here, we estimate it via simple kernel-density estimation based on the original sample pair $\{(x_0, p_0), (x_1, p_1)\}$. First, we rewrite $p(y|\tilde{x})$ by using Bayes's theorem as the following equation:

$$p(y = j|\tilde{x}) = \frac{\pi_j p(\tilde{x}|y = j)}{p(\tilde{x})}, \tag{7}$$

where $\pi_j$ denotes $p(y = j)$ and $j \in \{1, 2, ..., C\}$. Then, intead of directly estimating $p(y|\tilde{x})$, we estimate both $p(\tilde{x})$ and $p(\tilde{x}|y)$ by using a kernel function $k$ as

$$p(\tilde{x}) = \sum_{i \in \{0,1\}} \frac{1}{2} p(\tilde{x}|y = y_i), \; p(\tilde{x}|y) = \begin{cases} k(\tilde{x} - x_0) & \text{if } y = y_0, \\ k(\tilde{x} - x_1) & \text{if } y = y_1, \\ 0 & \text{otherwise.} \end{cases} \tag{8}$$

Since we only use the two samples, $(x_0, p_0)$ and $(x_1, p_1)$, for this estimation, $\pi_j$ is set to $1/2$ if $j \in \{y_0, y_1\}$ and 0 otherwise. By substituting Eqs. (8) into Eq. (7), we obtain the following equation:

$$p(y|\tilde{x}) = \begin{cases} \frac{k(\tilde{x}-x_0)}{\sum_{i \in \{0,1\}} k(\tilde{x}-x_i)} & \text{if } y = y_0, \\ \frac{k(\tilde{x}-x_1)}{\sum_{i \in \{0,1\}} k(\tilde{x}-x_i)} & \text{if } y = y_1, \\ 0 & \text{otherwise.} \end{cases} \tag{9}$$

To make $\tilde{p}$ represent this class posterior probability, we need to set the interpolation ratio $\lambda_b$ in Eq. (6) as the following equation:

$$\lambda_b = p(y = y_0|\tilde{x}) = \frac{k(\tilde{x} - x_0)}{\sum_{i \in \{0,1\}} k(\tilde{x} - x_i)}. \tag{10}$$

Once we generate the interpolated data, we can determine the labels of the interpolated data by using Eq. (10). Obviously, $\lambda_b$ is not necessarily equal to $\lambda_a$, which is different from Mixup.

### 2.2.2 How to set the ratio for the data interpolation

Since we have already formulated $p(\tilde{x})$, we have to carefully set the ratio for the data interpolation so that the resulting interpolated samples follow this distribution. Specifically, we cannot simply use the beta distribution to sample $\lambda_a$, which is used in Mixup, and need to specify an appropriate distribution $p(\lambda_a)$ to guarantee the interpolated data follow $p(\tilde{x})$ shown in Eq. (8).

By using Eq. (5) and Eq. (8), we can formulate $p(\lambda_a)$ as the following equation:

$$p(\lambda_a) = p(\tilde{x}) \left| \frac{d\tilde{x}}{d\lambda_a} \right| = |x_0 - x_1| \sum_{i \in \{0,1\}} \frac{1}{2} k(\tilde{x} - x_i). \tag{11}$$

Since the kernel function $k$ is defined in the $x$-space, it is hard to directly sample $\lambda_a$ from the distribution shown in the right-hand side of Eq. (11). To re-formulate this distribution to make it easy to sample, we define another kernel function in the $\lambda_a$-space as

$$k'(\lambda_a) = |x_0 - x_1| \, k(\lambda_a(x_0 - x_1)). \tag{12}$$

By using this new kernel function, we can rewrite $p(\lambda_a)$ in Eq. (11) and also $\lambda_b$ in Eq. (10) as follows:

$$p(\lambda_a) = \sum_{i \in \{0,1\}} \frac{1}{2} k'(\lambda_a - (1-i)), \qquad (13)$$

$$\lambda_b = \frac{k'(\lambda_a - 1)}{\sum_{i \in \{0,1\}} k'(\lambda_a - (1-i))}. \qquad (14)$$

This formulation enables us to easily sample $\lambda_a$ and to determine its corresponding $\lambda_b$. Note that we need to truncate $p(\lambda_a)$ when sampling $\lambda_a$ to guarantee that the sampled $\lambda_a$ is in the range of $[0, 1]$. The kernel function $k'$ should be set manually before training. It corresponds to a hyper-parameter setting of the beta distributon in case of Mixup.

## 3 EXPERIMENTS

### 3.1 CONFIDENCE ESTIMATION

We first conducted experiments to confirm how much MixConf contributes to improve the confidence calibration especially when the number of training data is small. We used object recognition datasets (CIFAR-10 and CIFAR-100 (Krizhevsky, 2009)) and a fashion-product recognition dataset (Fashion MNIST (Xiao et al., 2017)) as in the previous study (Thulasidasan et al., 2019). The number of training data is 50,000 for the CIFAR-10 / CIFAR-100 and 60,000 for the Fashion MNIST. To make a small-scale training dataset, we randomly chose a subset of the original training dataset while keeping class priors unchanged from the original. Using this dataset, we trained ResNet-18 (He et al., 2016) by the Adam optimizer (Kingma & Ba, 2015). After training, we measured the confidence calibration of the trained models at test data by the Expected Calibration Error (ECE) (Naeini et al., 2015):

$$\text{ECE} = \sum_{m=1}^{M} \frac{|B_m|}{N} \left| \text{acc}(B_m) - \text{conf}(B_m) \right|, \qquad (15)$$

where $B_m$ is a set of samples whose confidence fall into $m$-th bin, $\text{acc}(B_m)$ represents an averaged accuracy over the samples in $B_m$ calculated by $|B_m|^{-1} \sum_{i \in B_m} 1(\hat{y}_i = y_i)$, and $\text{conf}(B_m)$ represents an averaged confidence in $B_m$ calculated by $|B_m|^{-1} \sum_{i \in B_m} \hat{c}_i$. We split the original test data into two datasets, namely 500 samples for validation and the others for testing. Note that the validation was conducted by evaluating the prediction accuracy of the trained model on the validation data, not by the ECE. For each setting, we conducted the experiments five times with random generation of the training dataset and initialization of the model, and its averaged performance will be reported.

For our MixConf, we tried the Gaussian kernel and the triangular kernel, and we call the former one MixConf-G and the latter one MixConf-T. The width of the kernel, which is a hyper-parameter of MixConf, is tuned via the validation. For comparison, we also trained the models without Mixup as a baseline method and those with Mixup. Mixup has its hyper-parameter $\alpha$ to determine $p(\lambda_a) = Beta(\alpha, \alpha)$, which is also tuned in the experiment.

Figure 3 shows the ECE of the trained models. The horizontal axis represents the proportion of the original training data used for training, and the vertical axis represents the ECE of the model trained with the corresponding training dataset. In all methods, the ECE increases when the number of training data gets small, which indicates that the over-confidence problem of DNNs gets bigger in case of the small-scale training dataset. As reported in (Thulasidasan et al., 2019), Mixup substantially reduces the ECE compared with the baseline method, but its ECE still increases to some extent when the training dataset becomes small-scale. MixConf-G succeeds in suppressing such increase and achieves lower ECE in all cases. The performance of MixConf-T is not so good as that of MixConf-G especially in case of CIFAR-10/100. Since the actual width of the kernel function in the data space gets small according to the increase of the training data due to smaller $|x_0 - x_1|$ (see Eq. (12)), the difference between MixConf and Mixup becomes small, which results in similar performance of these methods when the number of the training data is large. Through the almost all settings, MixConf-G with $\sigma = 0.4$ performs best. Therefore, we used it in our SSL method in the experiments shown in the next section.

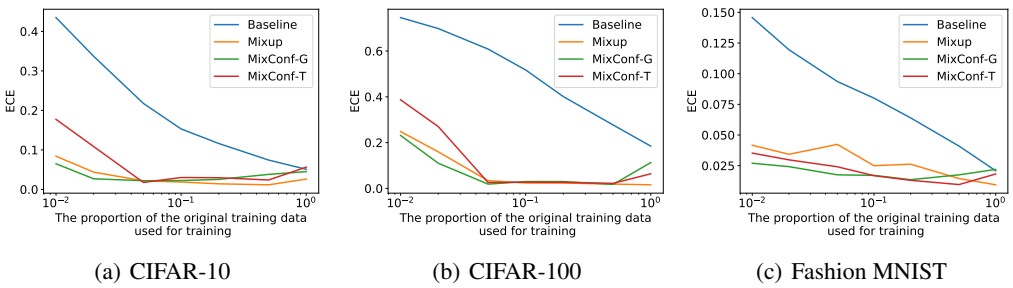

(a) CIFAR-10          (b) CIFAR-100          (c) Fashion MNIST

Figure 3: The Expected Calibration Error (ECE) of the trained models.

## 3.2 SEMI-SUPERVISED LEARNING

To validate the advantage of our SSL method, we conducted experiments with popular benchmark datasets: CIFAR-10 and SVHN dataset (Netzer et al., 2011). We randomly selected 1,000 or 4,000 samples from the original training data and used them as labeled training data while using the remaining data as unlabeled ones. Following the standard setup (Oliver et al., 2018), we used the WideResNet-28 model. We trained this model by using our method with the Adam optimizer and evaluated models using an exponential moving average of their parameters as in (Berthelot et al., 2019). The number of iterations for training is set to 400,000. The hyper-parameters $(\lambda_U, c_{thr})$ in our method are set to $(2, 0.8)$ for CIFAR-10 and $(3, 0.6)$ for SVHN dataset, unless otherwise noted. We report the averaged error rate as well as the standard deviation over five runs with random selection of the labeled data and random initialization of the model. We compared the performance of our method with those of several recent SSL methods, specifically, Virtual Adversarial Training (VAT) (Miyato et al., 2018), Interpolation Consistency Training (ICT) (Verma et al., 2019), Mix-Match (Berthelot et al., 2019), Pseudo-labeling (Lee, 2013), and Hermite-SaaS (Lokhande et al., 2020). Note that the former three methods utilize soft pseudo-labels, while the latter two use hard ones. We did not include ReMixMatch (Berthelot et al., 2020) in this comparison, because it adopts an optimization of data-augmentation policy that heavily utilizes domain knowledge about the classification task, which is not used in the other methods including ours.

Table 1 shows the test error rates achieved by the SSL methods for each setting. For CIFAR-10, our method has achieved the lowest error rates in both settings. Moreover, our method has shown relatively fast convergence; for example, in case of 1,000 labels, the number of iterations to reach $7.75\%$ in our method was around 160,000, while that in MixMatch is about 1,000,000 as reported in (Berthelot et al., 2019). Lokhande et al. (2020) have reported much faster convergence, but our method outperforms their method in terms of the test error rates by a significant margin. For SVHN dataset, our method has shown competitive performance compared with that of the other state-of-the-art methods.

We also conducted an ablation study and investigated performance sensitivity to the hyper-parameters using CIFAR-10 with 1,000 labels. The results are shown in Table 2. When we set $K = 1$ or use Mixup instead of MixConf, the error rate substantially increases, which indicates that it is important to accurately estimate the confidence of the pseduo labels in our method. On the other hand, the role of the small-loss selection is relatively small, but it shows distinct improvement. Decreasing the value of $\lambda_U$ leads to degraded performance, because it tends to induce overfitting to small-scale labeled data. However, if we set too large value to $\lambda_U$, the training often diverges due to overly relying on pseudo labels. Therefore, we have to carefully set $\lambda_U$ as large as possible within a range in which the model is stably trained. The confidence threshold $c_{thr}$ is also important; the test error rate varies according to the value of $c_{thr}$ as shown in Fig. 4. Considering to accept pseudo-labeled data as much as possible, smaller $c_{thr}$ is preferred, but too small $c_{thr}$ substantially degrade the performance due to increasing a risk of selecting incorrectly pseudo-labeled data to calculate the loss. We empirically found that, when we gradually decrease the value of $c_{thr}$, the training loss of the trained model drastically decreases at a little smaller $c_{thr}$ than the optimal value as shown by a red line in Fig. 4. This behavior should provide a hint for appropriately setting $c_{thr}$.

Table 1: Experimental results on CIFAR-10 and SVHN dataset.

| | Method | CIFAR-10 | | SVHN | |
|---|---|---|---|---|---|
| | | 1,000 labels | 4,000 labels | 1,000 labels | 4,000 labels |
| Soft pseudo-labels | VAT (Miyato et al., 2018) | 18.68±0.40 | 11.05±0.31 | 5.98±0.21 | 4.20±0.15 |
| | ICT (Verma et al., 2019) | - | 7.66±0.17 | 3.53±0.07 | - |
| | MixMatch (Berthelot et al., 2019) | 7.75±0.32 | 6.24±0.06 | **3.27**±0.31 | **2.89**±0.06 |
| Hard pseudo-labels | Pseudo-labeling (Lee, 2013) | 31.53±0.98 | 17.41±0.37 | 10.19±0.41 | 5.71±0.07 |
| | Hermite-SaaS (Lokhande et al., 2020) | 20.77 | 10.65 | 3.57±0.04 | - |
| | Our method | **7.13**±0.08 | **5.81**±0.12 | 3.63±0.12 | 3.23±0.06 |

Table 2: Ablation study. We used CIFAR-10 dataset with 1,000 labels.

| Ablation | Error rate |
|---|---|
| Proposed method ($K = 4$, $\lambda_U = 2$, $c_{thr} = 0.80$) | 7.13 |
| With $K = 1$ | 8.33 |
| With $\lambda_U = 1$ | 7.89 |
| With Mixup instead of MixConf | 7.73 |
| Without the small-loss selection | 7.39 |

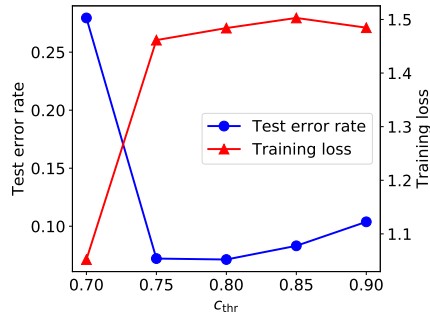

Figure 4: Test error rates and training loss of the trained model in our method.

## 4 CONCLUSION

In this paper, we presented a novel SSL method that adopts selective training with pseudo labels. In our method, we explicitly select the pseudo-labeled data that correspond to relatively small loss after the data augmentation is applied, and only the selected data are used to train the model, which leads to effectively preventing the model from training with incorrectly pseudo-labeled data. We estimate the confidence of the pseudo labels when generating them and use it to determine the number of the samples to be selected as well as to discard inaccurate pseudo labels by thresholding. We also proposed MixConf, which is the data augmentation method that enables us to train more confidence-calibrated models even in case of small-scale training data. Experimental results have shown that our SSL method performs on par or better than the state-of-the-art methods thanks to the selective training and MixConf.

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
