# OpenReview forum: "Semi-supervised learning by selective training with pseudo labels via confidence estimation"
_ICLR.cc/2021/Conference — Reject_

### Official Review · AnonReviewer1 · 2020-10-25
**Insufficient novelty**

**Rating:** 4
**Confidence:** 3

**Review:**

This paper presents a pseudo-labeling-based method for semi-supervised leanring. The proposed method consists of (1) pseudo-labeled data selection based on prediction confidence for efficient training and (2) a data augmentation method named mixconf, which is a modification of mixup.

Overall, the paper is well written, but i can't find much novelty of the proposed method. For the part (1), data selection for semi-supervised learning is a quite well-known idea. For the part (2),  It's just a very simple and obvious modification of the existing data augmentation method (which may lead to marginal performance improvement. it's effectiveness should be additionally demonstrated using other benchmark datasets).

Also I think the authors should perform more experimental investigations to support their idea. For example,
1) The main assumption the authors made is "the loss calculated with incorrect labels tends to sensitivey increase when the data is augmented". I think they can give examples about that on the experimented datasets to show that their assumption is more likely.
2) it seems the proposed method involves so many hyperparameters and it seems its outcome is sensitive to the settings of the hyperparameters, while how they tuned the hyperparameters is unclear, hidden at the back. Table 2 and Figure 4 (results on CIFAR-10 dataset) show that the performance of the proposed method is considerably affected by the hyperparameters. One thing they must do is to carry out ablation study on the SVHN dataset and if possible, other new datasets.
3) For semi-supervised learning, there is no reason to prefer pseudo-labeling approach to others (e.g. semi-supervised VAE, ladder nets). Because the other methods have been very effective in semi-supervised learning, they can be additionally compared with the proposed one.
4) One benefit of the proposed method the authors mentionedis its computational efficiency (by selecting highly confident pseudo-labeled data). This could be wrong if it involves more training iterations due to the other components of the proposed method (such as training configurations and data augmentation stuffs, etc.). To clearly demonstrate if the proposed method is actually computationally efficient, the authors should provide the comparison of the training time each method required.

---

### Official Review · AnonReviewer3 · 2020-10-27
**This paper proposes a semi-supervised method by assigning pseudo-labels with confidence estimation.**

**Rating:** 6
**Confidence:** 3

**Review:**

In this paper, a novel semi-supervised learning (SSL) method that adopts selective training with pseudo labels is proposed based on confidence estimation. The confidence is used not only for screening candidates of pseudo-labeled data to be selected but also for automatically deciding how many pseudo-labeled data should be selected within a mini-batch.
[1] In the data selection, it selects the data corresponding to relatively small loss after applying data augmentation. What is the advantage of the proposed method compared with the conditional GAN.
[2] How to balance the number of samples that are generated for training in each class.
[3] In the proposed method, it needs to select the  samples with small loss. It may be time-consuming, since the loss of each samples should be estimated. Meanwhile, the Gaussian kernel is adopted in the proposed method. As we know, the Gaussian kernel is very cost on both memory and time with the number of samples increasing. Hence, how about the overhead of the proposed method? The overhead of the proposed method analysis may be very important to further inspect the proposed method.
[4] Whether the proposed method is still effective if the number of the labeled training data is very limited.

---

### Official Review · AnonReviewer4 · 2020-10-27
**Some unclarities, partially but not only due to lack of background knowledge**

**Rating:** 5
**Confidence:** 2

**Review:**

This paper proposes a new method to perform semi-supervised learning by self-annotating data at each iteration, simply choosing to do so by selecting a subset of instances to which are assigned a pseudo-label, those chosen instances being the one for which a loss is minimal once data have been mixed through a MixConf procedure.

While familiar with semi-supervised learning and label propagation, I must say I had some difficulties following part of the paper, partly because I was not familiar with the initial mixing methods and their justifications. This means that I would like some more justifications of why this is a sound method to select items to label. The results of the experiments are also a bit disappointing, as out of two data sets, the provided method seems to give better results on only one data set, albeit one of its advantage is to require less iterations (by a magnitude of 10) than other soft-label based methods to reach similar performances (albeit the practical advantage of such a speed up heavily depends on the time taken by an iteration).

Here are some clarifications that could be helpful to better understand the paper:

* In equations (2) and (3), I did not find the information (in the experimental setting) of how were fixed B_U and B_L, that in turns allow to select n_L and n_U data. A relation between B_U and B_L is given, but that still requires to settle one of the number for n_L (and n_U) to be well defined.

* In equation (4), how is it that the loss is between an instance and a probability distribution over classes? Or is \tilde{x}^L_{s[i]} the class?

* Section 2: what theoretical argument have we that the proposed approach will provide well-calibrated outputs? The method actually produces a bunch of bi-modal distributions whose posterior density and input density are estimated by non-parametric kernels over two data. Also, it may well be that the average \lambda_a x_0 + (1-\lambda_a) x_1 is a non-observable point (in this case, image), hence that it has actually probability zero? The experiments seem to show that it works reasonably well on the three selected data sets (as did MixMatch), but do we have formal arguments indicating that this approach would provide well-calibrated outputs in general?

* Figure 3 and calibration curves: beyond the baseline, all three methods seem to be on par regarding calibration problems, so it is unclear whether MixConf actually brings any water to the mill of semi-supervised learning, data augmentation or calibration? Also, how can we explain that more data actually degrade the calibration results, as one would expect calibration to improve with data? Would it be possible to show calibration curves?

* Semi-supervised learning: basically, the proposed approach consists in two different bricks, the calibration/mixing approach, and the selection/labelling approach. In the comparison with MixMatch, both aspects are changed. What happens when we apply the same selection/labelling approach to MixMatch? It would probably give us the same speed up, but with a different mixing techniques. I guess what I would like to see is an ablation study between the present paper and the MixMatch approach, as right now the real added value of MixConf is unclear to me (both in Figure 3 and in Table 1).

* There are also some typos remaining (in a small amount), so a final read would probably be helpful. e.g.: "not that during training".

---

### Official Review · AnonReviewer2 · 2020-10-29
**Semi-supervised learning by selective training with pseudo labels via confidence estimation**

**Rating:** 5
**Confidence:** 4

**Review:**

The paper uses selective training with pseudo labels. Specifically, the method selects the pseudo-labeled data associated  with small loss after performing  the data augmentation, and then uses  the selected data for training the model. Here, the model computes the confidence of the pseudo labels and then puts a threshold to determine  the number of the selected samples and ignore inaccurate pseudo labels. Moreover, MixConf, a variation of mixup, for data augmentation is proposed  to train a more confidence calibrated model. Finally, experimental results on the standard datasets  show the effectiveness of the proposed model compared to SOA SSL methods.


The paper is well-written, and experiments are sufficient to some extent. However, the technical novelty of the work is marginal. This is because,  treating unlabeled data differently for SSL has been recently investigated in the following work [1], and this work can be readily modified to perform in a selective training fashion (e.g., select sample or not during the training). Moreover, the work is very similar to the SSL models that use MixUp for data augmentation such as Mixmatch,  ReMixMatch and  [2]

[1]- https://arxiv.org/abs/2007.01293

[2]- Arazo E, Ortego D, Albert P, O’Connor NE, McGuinness K. Pseudo-labeling and confirmation bias in deep semi-supervised learning. In2020 International Joint Conference on Neural Networks (IJCNN) 2020 Jul 19 (pp. 1-8). IEEE.

-Finally, extracting confident examples with small losses, or  selecting noisy/out of distribution samples based on loss or output of the softmax is explored previously in the literature for  sample selection and training deep models with noisy labels, examples of this are provided as follows:

[3] Han B, Yao Q, Yu X, Niu G, Xu M, Hu W, Tsang I, Sugiyama M. Co-teaching: Robust training of deep neural networks with extremely noisy labels. InAdvances in neural information processing systems 2018 (pp. 8527-8537).

[4] Xingrui Yu, Bo Han, Jiangchao Yao, Gang Niu, Ivor W Tsang, and Masashi Sugiyama. How
does disagreement benefit co-teaching? In ICML, 2019.

[5] Xiaobo Wang, Shuo Wang, Jun Wang, Hailin Shi, and Tao Mei. Co-mining: Deep face recognition with noisy labels. In ICCV, pages 9358–9367, 2019.

[6] Chen Y, Zhu X, Li W, Gong S. Semi-Supervised Learning under Class Distribution Mismatch. InAAAI 2020 (pp. 3569-3576).


-I think Tuning c_{thr} needs more  attention and it highly depends on many factors such as the datasets, number of initially labeled data for training, and etc. Considering these, it may affect the performance of the model differently. So I encourage the authors to elaborate more on tuning this parameter.

-The authors may add the performance of the model in a fully supervised fashion, it  is good to know the lower and upper bound of the network's  performance. Since  the results of the proposed model are marginally better than the Mixmatch, it is a bit difficult to compare these methods, specially when  the  training set-up is different.  In such a case, the gap between the lower bond and the reported results might be better for a fair comparison.

-Could you please provide some discussion on the similarity and differences between your methods and the works discussed above?

---

### Decision · Program_Chairs · 2021-01-07
**Final Decision**

**Decision:**

Reject

**Comment:**

The paper proposes an approach that generates pseudo-labels along with confidence to help semi-supervised learning. Then, selected pseudo-labels are used to update the model. Moreover, the authors include a variation of mixup for data augmentation to train a more calibrated model. Experimental results justify the validity of the proposed approach.

Several reviewers believe that the paper is somewhat well-written. The main concern is on the novelty of the work. In particular, many works have discussed selected treatment of unlabeled data, data augmentation for semi-supervised learning, and label confidence estimation. Those works deserve more discussions/comparisons. The paper can also be improved with deeper experimental studies that better justify the main assumption and merits of the proposed approach.